# Genetic study links components of the autonomous nervous system to heart-rate profile during exercise

Niek Verweij[1], Yordi J. van de Vegte[1] & Pim van der Harst [1,2,3]

Heart rate (HR) responds to exercise by increasing during exercise and recovering after exercise. As such, HR is an important predictor of mortality that researchers believe is modulated by the autonomic nervous system. However, the mechanistic basis underlying inter-individual differences has yet to be explained. Here, we perform a large-scale genome-wide analysis of HR increase and HR recovery in 58,818 UK Biobank individuals. Twenty-five independent SNPs in 23 loci are identified to be associated ($p < 8.3 \times 10^{-9}$) with HR increase or HR recovery. A total of 36 candidate causal genes are prioritized that are enriched for pathways related to neuron biology. No evidence is found of a causal relationship with mortality or cardiovascular diseases. However, a nominal association with parental lifespan requires further study. In conclusion, the findings provide new biological and clinical insight into the mechanistic underpinnings of HR response to exercise. The results also underscore the role of the autonomous nervous system in HR recovery.

[1] University of Groningen, University Medical Center Groningen, Department of Cardiology, 9700 RB Groningen, The Netherlands. [2] University of Groningen, University Medical Center Groningen, Department of Genetics, 9700 RB Groningen, The Netherlands. [3] Durrer Center for Cardiogenetic Research, ICIN - Netherlands Heart Institute, 3511GC Utrecht, The Netherlands. Correspondence and requests for materials should be addressed to N.V. (email: mail@niekverweij.com)

Physical activity places an increased demand on a person's cardiovascular capabilities. This activity relies heavily on cardiovascular health and regulation by the autonomic nervous system[1]. Electrocardiograms (ECGs) of exercise tests are used to determine cardiac fitness and function; they offer unique insights into cardiac physiology compared to ECGs performed on people at rest. The first data linking electrocardiographic changes in response to exercise with mortality was presented in 1975. This data indicated that a low-peak heart-rate (HR) response during exercise was associated with an increased risk of cardiac death[2]. It is now well accepted that chronotropic incompetence confers a worse prognosis for cardiac mortality and events[3]. Increased HR during exercise and HR recovery after exercise is specifically associated with sudden cardiac death and all-cause mortality in healthy individuals[4–6]. Increased HR during these periods has been observed in coronary and heart failure patients regardless of β-blocker usage[7–9]. The majority of previous studies focused on HR recovery at 60 s, which is strongly heritable (at around 60%[10]). The hypothesis linking HR recovery to mortality arose from work that associated components of the autonomic nervous system with sudden cardiac death[11], as well as studies of decreased vagal activity[12,13]. McCrory et al[14]. recently expanded on this topic by adding additional evidence linking baroreceptor dysfunction with mortality. The study also identified HR recovery in the first ten seconds after an orthostatic challenge to be most predictive of mortality. The cardiovascular system's immediate response to exercise is an increased HR that is attributable to a decrease in vagal tone followed by an increase in sympathetic outflow and, to some extent, circulating hormones[15]. The mechanism to reduce HR after exercise follows the inverse mechanism, a gradient of parasympathetic nervous system reactivation and sympathetic withdrawal[15]. The effect of this reactivation is believed to be strongest in the first 30 s after the termination of exercise[16]. However, the exact molecular mechanisms underlying inter-individual differences in HR response to exercise, as defined by HR increase and HR recovery, are unknown.

The UK Biobank includes a sub-cohort of 96,567 participants who were invited for electrocardiographic exercise testing. This cohort enabled the possibility to carry out in-depth genetic analyses of HR response to exercise for the first time. The goals of the current study study are to (1) provide (shared) genetic heritability estimates among variables of the HR profile during exercise; (2) identify genetic variants and the underlying candidate causal genes associated with HR increase and HR recovery at 10, 20, 30, 40, and 50 s; and (3) obtain insights into pleiotropy and the clinical consequences of HR increase and HR recovery. We revealed extensive genetic pleiotropy among phenotypes of the HR profile during exercise and find 23 significantly associated genetic loci. No evidence for a causal relationship was found between HR increase or HR recovery and parental lifespan or disease outcomes. Nevertheless, the genetic loci provide support for the hypothesis that the autonomous nervous system is a major player in regulating HR recovery. Collectively, the results improve our understanding of HR regulation in response to exercise from a genetics perspective.

## Results

**Genetics of the HR profile during exercise.** Participants from the UK Biobank exercised for ~350 (±44.9) seconds; the mean duration of the recovery phase was 52.6 (±1.7) seconds. Overall characteristics are presented in Supplementary Table 1. All HR phenotypes were normally distributed prior to rank-based inverse normal transformation.

To gain insights into the correlations between phenotypes of the HR profile during exercise, we first performed heritability analyses and genetic correlations across nine HR phenotypes: the increase in HR from resting level to peak exercise level (HR increase) and the decrease in rate from peak exercise level to the level 10, 20, 30, 40, and 50 s after termination of exercise (HRR10-HRR50). Resting HR and HR variability as defined by SDNN and RMSSD were included for comparison. The highest heritability estimates were observed for HR recovery and HR increase ($h_{2gSNP}$ = 0.22). HR variability was much less heritable ($h_{2gSNP}$ = 0.12 and 0.14 for SDNN and RMSSD) based on SNP heritability estimates by BOLT-REML (Fig. 1). All of the HR variables were highly correlated with each other (Fig. 1), though HR recovery and HR increase were more strongly correlated with each other ($r$ = 0.6–0.9) than with HR variability ($r$ = 0.42–0.6) or resting HR ($r$ = −0.18−−0.37). The genotypic correlations were slightly higher compared to the phenotypic correlations. All of the heritability and correlation estimates were highly significant ($p <$ $1 \times 10^{-8}$).

Genome-wide association analyses were conducted for HR increase and HRR10-HRR50. Twenty-three genomic loci defined by 1MB at either side of the highest associated SNP were significant, $p < 8.3 \times 10^{-9}$, and are summarized in Table 1; Fig. 2 shows the Manhattan plot, visualizing the distribution of genetic variants in the genome and Supplementary Data 1 provides a more expanded summary of the results per individual HR-phenotype. LD score regression[17] on the genome-wide summary statistics yielded intercepts that ranged between 1.004 (HRR10) and 1.014 (HRR20), indicating that any inflation of genomic control can be attributed to polygenicity rather than sources like residual population stratification (Supplementary Fig. 1). Two additional independent signals in two loci on chromosome 2 and 5 were confirmed by conditional analyses (Supplementary Table 2). rs6488162 in *SYT10* was the most significant genetic variant for all phenotypes ($p = 3.1 \times 10^{-30}$ for HR increase, to $p$ $= 5.3 \times 10^{-66}$ for HRR10). Results of the sensitivity analyses are presented in Supplementary Data 2 and indicate that the SNP associations were not biased by participants receiving medication or having heart disease diagnoses. Supplementary Fig. 2 illustrates the regional association plots of each locus.

**Insights into biology.** A total of 36 candidate causal genes were identified at the 23 loci. Twenty-seven genes were prioritized based on proximity to the sentinel SNP, three genes were prioritized by coding variants in LD of $R^2 > 0.8$ with a sentinel SNP (summarized in Supplementary Data 3), 10 genes were prioritized by eQTL analyses (tissue-specific eQTLs are shown in Supplementary Data 4), and 11 genes were prioritized by long-range interaction analyses in Hi-C data (listed in Supplementary Table 3 and visualized in Supplementary Fig. 3). Multiple lines of evidence may have prioritized a gene (indicated by the candidate causal gene superscripts in Table 1), further prioritizing the most likely candidate causal genes and mechanisms at each locus.

Pathway analyses were attempted with 'DEPICT' using default settings (which uses all SNPs $p < 1 \times 10^{-5}$), a tool that can prioritize genes, pathways, and tissues by using the genomic region surrounding SNPs as input (please see Pers et al.[18] for a detailed description of the methods). However, no significant pathways or tissues were identified after correcting for multiple testing. Instead, GeneNetwork[19] (http://129.125.135.180:8080/GeneNetwork/pathway.html) was used; this method employed the same underlying co-expression dataset (based on GEO data), but allowed only the 36 candidate causal genes as input. The candidate causal genes were enriched for terms related to neurons and axons ('axon guidance', 'neuron recognition' 'peripheral nervous system neuron development', and 'synapse') and gap junctions ('adherents junction organization' and 'gap junction'),

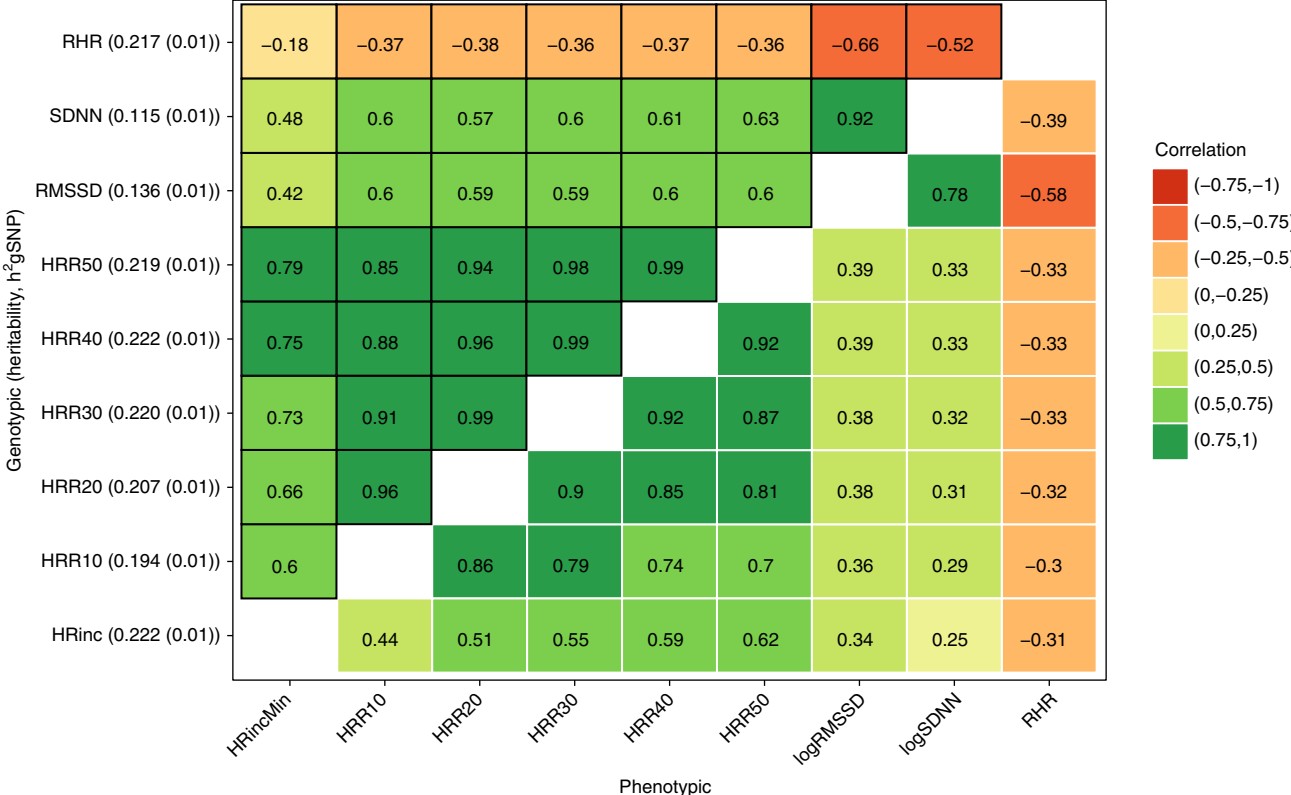

**Fig. 1** Shared genetic correlations and heritability estimates of the HR profile during exercise. Genetic correlations (shared heritability), are shown above the diagonal, phenotypically observed correlations are below the diagonal. Heritability estimates (and standard errors) of each trait are between brackets at the y-axis. All of the estimates shown here were highly significant ($p < 10^{-8}$). Correlations are based on the residual variance after adjustments for age, sex and BMI, exercise-specific variables and genetic-specific variables (only for the genetic correlations)

### Table 1 List of 25 genome-wide significant SNPs in 23 loci that are associated with HR increase or HR recovery

| CHR | SNPs | Position (hg19) | EA(Freq)/NEA | Beta | SE | p | Candidate gene | Primary Trait |
|---|---|---|---|---|---|---|---|---|
| 1 | rs11589125 | 31894396 | T(0.06)/C | 0.075 | 0.013 | $6.60 \times 10^{-09}$ | $SERINC2^{n,c}$ | HRR50 |
| 1 | rs272564 | 45012273 | A(0.71)/C | 0.046 | 0.007 | $1.40 \times 10^{-12}$ | $RNF220^{n,h}$ | HRR50 |
| 1 | rs61765646 | 72723211 | A(0.19)/T | 0.056 | 0.008 | $1.10 \times 10^{-13}$ | $NEGR1^{n}$ | HRR10 |
| 2 | rs1899492 | 60000304 | T(0.47)/C | 0.040 | 0.006 | $1.70 \times 10^{-11}$ | Gene desert | HRR40 |
| 2 | rs17362588 | 179721046 | G(0.92)/A | 0.062 | 0.011 | $3.10 \times 10^{-09}$ | $CCDC141^{n,c},TTN^{h}$ | HRR10 |
| 2 | rs35596070 | 179759692 | C(0.86)/A | 0.060 | 0.008 | $4.20 \times 10^{-13}$ | $CCDC141^{n,c},TTN^{h}$ | HRR10 |
| 3 | rs73043051 | 18883863 | C(0.22)/T | 0.041 | 0.007 | $7.80 \times 10^{-09}$ | $KCNH8^{n}$ | HRR50 |
| 3 | rs34310778 | 74783408 | C(0.43)/T | 0.036 | 0.006 | $1.00 \times 10^{-09}$ | $CNTN3^{n,e}$ | HRR30 |
| 5 | rs4836027 | 121866990 | T(0.68)/C | 0.050 | 0.006 | $1.70 \times 10^{-15}$ | $SNCAIP^{n},PRDM6^{n,h}$ | HRinc |
| 5 | rs151283 | 122446619 | C(0.72)/A | 0.042 | 0.007 | $1.60 \times 10^{-10}$ | $PRDM6^{nh}$ | HRR50 |
| 6 | rs2224202 | 102053814 | A(0.19)/G | 0.043 | 0.007 | $5.80 \times 10^{-09}$ | $GRIK2^{n,h}$ | HRR20 |
| 7 | rs2158712 | 26582733 | A(0.52)/T | 0.045 | 0.006 | $2.80 \times 10^{-13}$ | $SKAP^{n,h}$ | HRR10 |
| 7 | rs180238 | 93550447 | T(0.65)/C | 0.043 | 0.006 | $2.20 \times 10^{-12}$ | $GNG11^{n},GNGT1^{n,e},TFPI2^{e}$ | HRR40 |
| 7 | rs3757868 | 100482720 | G(0.82)/A | 0.077 | 0.008 | $5.60 \times 10^{-24}$ | $SRRT^{n,e},ACHE^{n,e},TRIP6^{e},C7orf43^{n,e},UFSP1^{n}$ | HRR40 |
| 7 | rs1997571 | 116198621 | A(0.58)/G | 0.042 | 0.006 | $1.70 \times 10^{-12}$ | $CAV1^{n,h},CAV2^{n,e,h}$ | HRR20 |
| 7 | rs17168815 | 136624621 | G(0.84)/T | 0.062 | 0.008 | $1.10 \times 10^{-14}$ | $CHRM2^{n}$ | HRR50 |
| 10 | rs7072737 | 102556175 | A(0.11)/G | 0.079 | 0.009 | $1.10 \times 10^{-17}$ | $PAX2^{n}$ | HRR40 |
| 11 | rs7130652 | 71984398 | T(0.07)/G | 0.076 | 0.011 | $3.40 \times 10^{-11}$ | $CLPB^{n,h},INPPL1^{n,e}$ | HRR10 |
| 12 | rs4963772 | 24758480 | A(0.15)/G | 0.090 | 0.008 | $1.20 \times 10^{-28}$ | $BCAT1^{n}$ | HRR40 |
| 12 | rs6488162 | 33593127 | C(0.58)/T | 0.103 | 0.006 | $2.60 \times 10^{-66}$ | $SYT10^{n},ALG10^{h}$ | HRR10 |
| 12 | rs61928421 | 116227249 | C(0.93)/T | 0.090 | 0.012 | $4.30 \times 10^{-15}$ | $MED13L^{n}$ | HRR40 |
| 14 | rs17180489 | 72885471 | C(0.14)/G | 0.055 | 0.008 | $2.50 \times 10^{-11}$ | $RGS6^{n,h}$ | HRinc |
| 15 | rs12906962 | 95312071 | T(0.67)/C | 0.048 | 0.006 | $2.70 \times 10^{-14}$ | $MCTP2^{n}$ | HRinc |
| 19 | rs12974440 | 5894386 | G(0.92)/A | 0.067 | 0.011 | $2.40 \times 10^{-10}$ | $FUT5^{n}, NDUFA11^{n,c}$ | HRR10 |
| 19 | rs12986417 | 30109533 | G(0.65)/A | 0.037 | 0.006 | $1.00 \times 10^{-09}$ | $POP4^{n},C19orf12^{h}$ | HRinc |

*HRinc* HR increase, *HRRx* HR recovery at x seconds, *CHR* Chromosome, *EA* effect allele, *NEA* Non-effect allele, *SE* Standard error

[n] nearest gene or any other gene in 10 kb
[c] coding variant gene
[e] eQTL gene
[h] Hi-C long-range interaction gene
More detailed information can be found in Supplementary Table 2 and 3

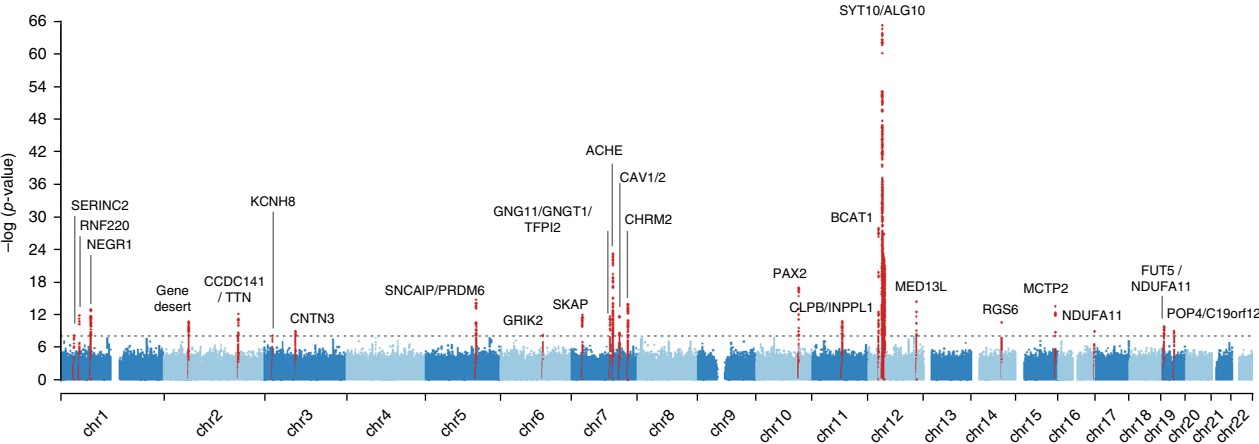

**Fig. 2** Manhattan plot of the GWAS of HR increase and recovery. The smallest *p*-values per SNP across all of the six studied traits are shown, as depicted on the *y*-axis, the *x*-axis shows their chromosomal (chr) positions. Red dots represent genome-wide significant loci ($p < 8.3 \times 10^{-9}$)

but also included 'catecholamine transport' and 'decreased dopamine level', among others (Supplementary Data 5). In a separate analysis based on the GTEx dataset, nerve tissue was also highly enriched compared to other tissues ($p < 0.01$, Supplementary Fig. 4).

**Insights into pleiotropy and clinical relevance.** To gain more insight into the potential mediating mechanisms at the genetic variant level, we looked up previously reported variants in the literature and the GWAS catalog. Of the 25 independent SNPs, eleven were in high LD ($R^2 > 0.6$) with previously identified SNPs for resting HR[20,21] or HR variability[22] (Supplementary Data 6). A wider search in the GWAS catalog revealed that SNPs in high LD ($R^2 > 0.6$) with rs61765646 (*NEGR1*) were previously associated with obesity; rs17362588 (*CCDC141/TTN* but not the independent SNP rs35596070) and rs12906962 (*MCTP2*) with diastolic blood pressure and rs7072737 (*PAX2*) with systolic blood pressure; and rs4963772 (*BCAT1*) with PR interval and rs1997571 (*CAV1*) with atrial fibrillation and PR interval (Supplementary Data 7). The majority, 15 of 23 loci, had not been previously identified in any GWAS. PhenoScanner[23] also indicated that HR-profile SNPs had pleiotropic effects with resting HR, atrial fibrillation, and other electrocardiographic traits (Supplementary Data 8).

Because a large portion of the loci had already been reported for their association with other HR phenotypes, we examined SNP association with the different HR traits in the current study, in order to disentangle the effects and identify SNPs that are primarily driven by HR increase and HR recovery. Linear regression analyses were performed across all associated SNPs and traits. Associations were adjusted for (1) resting HR; (2) resting HR and HR variability; and (3) resting HR, HR variability, and HR increase. Figure 3 illustrates that rs17362588 (*TTN/ CCDC141*) is primarily associated with resting HR and highlights the following loci for HR variability: rs17180489 (*RGS6*), rs12974440 (*NDUFA11*), and to a lesser degree rs180238 (*GNG11*, *GNGT1*, and *TFPI2*) because the associations with HR recovery and HR increase were diminished significantly with additional adjustments of SDNN and RMSSD. The analyses also indicated that rs272564 (*RNF220*), rs4836027 (*SNCAIP/PRDM6*), rs4963772 (*BCAT1*), rs12906962 (*MCTP2*), and rs12986417 (*POP4*) were primarily associated with HR increase following additional adjustments for HR increase. In total, 16 SNPs remained independently associated with HR recovery, including the most significant locus *SYT10*. The association statistics used to create Fig. 3 are available in Supplementary Data 9.

To explore potential clinical relevance, polygenic scores were constructed based on the genome-wide significant SNPs. The primary outcome variable was parental age as proxy for cardiovascular- and all-cause mortality[14,24]. The choice of disease outcomes and phenotypes was based on previous studies of HR response to exercise in relation to ventricular arrhythmia (sudden death)[4], atrial fibrillation[25], diabetes[26], cancer[27], blood pressure[14], reaction time, fluid intelligence[28], and depression[29] were selected based on their potential relationship with autonomic (dys) function in general. A higher polygenic score was consistently associated with increased parental age of death ($p = 5.5 \times 10^{-4}$). On further inspection, a significant association was found with father's age of death ($p = 5.5 \times 10^{-4}$, $N = 217,722$), but not with mother's age of death ($p = 0.202$, $N = 179,281$). The association with increased parental lifespan may hint at a potential association with all-cause mortality, which was not significant in the UK Biobank sample ($HR = 0.924(0.055)$, $p = 0.186$, $N_{cases} = 10,717$ (3.0%); cox survival model). However, statistical power was limited compared to parental age of death.

The polygenic score was also strongly associated with lower diastolic blood pressure ($p = 2.0 \times 10^{-25}$) and lower odds of hypertension ($p = 2.3 \times 10^{-4}$). The association with hypertension depended on diastolic blood pressure, as the association was abolished after diastolic blood pressure was introduced into the model. We hypothesized that the strong association of the polygenic score with diastolic blood pressure may be due to resting HR. This hypothesis was strengthened by the fact that (1) resting HR had strong genetic correlations with HR increase and HR recovery and (2) resting HR has a direct influence on diastolic blood pressure via peripheral resistance. After resting HR was adjusted for, the association with diastolic blood pressure was also abolished ($p = 0.126$). No convincing associations were found between the polygenic score and atrial fibrillation, coronary artery disease, ventricular arrhythmia, diabetes, or cancer. The results are presented in Table 2; Supplementary Data 10 describes trait-specific effects and Supplementary Fig. 5 describes statistical power. To facilitate future studies, complete summary statistics of all genetic variants and traits can be downloaded from https://doi.org/10.17632/tg5tvgm436.1.

## Discussion

In this large-scale genetic study of HR increase and HR recovery in 58,818 participants, we identified 25 independent genome-wide significant signals in 23 genetic loci. HR increase and HR recovery were found to be highly heritable, and the majority of the loci

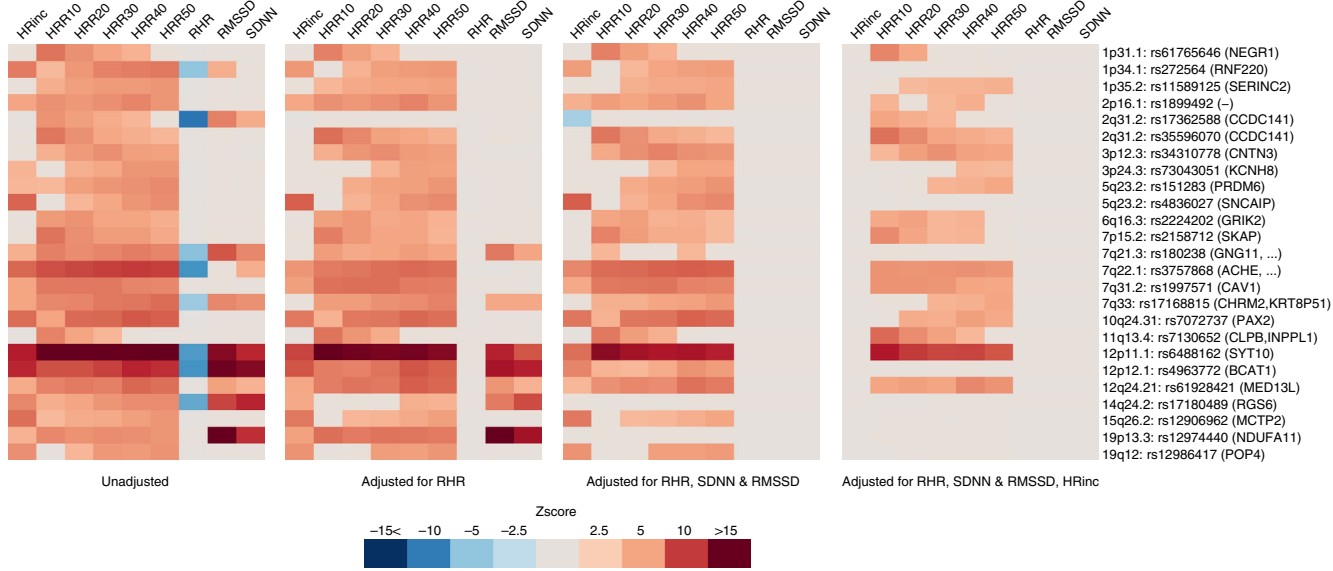

**Fig. 3** Pleiotropic effects of the 25 independent genetic signals on heart-rate (HR) phenotypes. Four heat plots depict Z-scores of each SNP association with resting HR (RHR), HR variability (RMSSD and SDNN), HR increase (HRinc), or HR recovery (HRR10- 50) in 1 univariate and 3 multivariable models (as described below each heat plot). Only Bonferonni $p < 0.05$ significant associations are shown, Z-scores were aligned to the allele that increases HR recovery. Nearby genes are shown between brackets

were independently associated with HR recovery. The polygenic score was not convincingly associated with mortality or disease.

The major finding was that a large number of candidate causal genes are involved in neuron biology, particularly at loci that are specific to HR recovery. This finding, together with our pathway analyses, provides a new line of evidence that the autonomic nervous system is a major player in the regulation of HR recovery. HR response to exercise, and HR recovery in particular, is largely dependent on parasympathetic reactivation and decrease of sympathetic activity in a gradual manner. These processes are orchestrated by neuronal signal transduction involving the brain (central command), periphery (chemoreflex, baroreflex, and exercise pressor reflex), adrenal medulla, and the actual nerves connecting these components[15].

The most significantly associated variant, rs6488162 in *SYT10*, encodes a $Ca^{2+}$ sensor Synaptotagmin 10 that triggers IGF-1 exocytosis, protecting neurons from degeneration[30]. Other loci include the *ACHE* gene, the function of which can be strongly linked to neuronal function as it encodes the enzyme that catalyzes the breakdown of acetylcholine. Neuronal Growth Regulator 1 (*NEGR1*) is essential for neuronal morphology. It has been demonstrated in in vitro and in vivo experiments that *NEGR1* overexpression and underexpression is closely associated with number of synapses by regulating neurite outgrowth and dendritic spine development[31]. *GRIK2* (also named GluR6) encodes a subunit of a kainite glutamate receptor that is broadly expressed in the central nervous system, where it plays a major role in nerve excitation[32]. *CHRM2* encodes the muscarinic acetylcholine receptor M2, which is the predominant form of muscarine cholinergic receptors in the heart. The receptor specifically initiates negative chronotropic and inotropic effects upon binding with acetylcholine released by the postganglionic parasympathetic nerves[33]. Hence, this gene corresponds well with results that rs17168815 (near CHRM2) is specifically associated with HR recovery. The gene *C19orf12* has an unknown function and is thought to encode a mitochondrial protein, several reports focus on mutations of *C19orf12* causing neurodegeneration[34]. The function of *MED13L* is also unclear, but is believed to encode a subunit that functions as a transcriptional coactivator for most

RNA polymerase II-transcribed genes. In zebrafish, *MED13L* knockdown causes abnormal effects on early migration of neural-crest cells, resulting in improper development of branchyal and pharyngeal arche, resembling key characteristics of *MED13L* mutations in humans[35]. *MED13L* mutations in humans are associated with intellectual disabilities, developmental delay, and craniofacial anomalies; these mutations also resemble other, more common, neurodevelopmental disorders[36]. *KCNH8* encodes a voltage-gated potassium channel that is primarily expressed in components of the human central nervous system[37] and is part of the Elk (ether-à-g-o go-like k) family of potassium channels that regulates neuronal excitation[37–39]. *CNTN3* (contactin-3) is a gene belonging to a group of glycosylphosphatidyl-anchored cell adhesion molecules that is found predominantly in neurons and is thought to be closely involved in the wiring of the nervous system[40,41]. In light of these findings, even *CCDC141* and not *TTN* (the main component of cardiac muscle) may be a plausible candidate gene, as *CCDC141* plays a crucial role in neuronal development[42]. Data of tissues that are relevant for the (para-)sympathetic nervous system of the heart are limited, which makes it difficult to dissect molecular mechanisms. Future research should pursue functional follow-up studies of the genetic loci presented here, to pinpoint causal variants, genes, and biological mechanisms underlying HR profile during exercise.

We observed that resting HR, HR variability, HR recovery, and HR increase were highly correlated with each other on the genetic and phenotypic levels. By jointly analyzing different HR traits rather than treating them as separate entities, as has been done traditionally, it was possible to obtain additional insights into the mechanistic basis of HR phenotypes. On the phenotypic level, insight into the genetic correlations helped us explain the strong association that was observed between the polygenic score and diastolic blood pressure. The association originated from resting HR; this finding is more plausible since resting HR is directly related to peripheral resistance. On the genetic variant level, we observed a large number of HR-recovery-specific SNPs to have neuronal genes as their candidate causal genes. Previous GWAS of resting HR have found genes predominantly enriched for terms related to cardiac structure[21], and a GWAS of HR variability

**Table 2 Association with clinical characteristics**

| Trait or disease | Sample size (% cases) | Effect size or odds ratio | se / 95%CI | p-value |
|---|---|---|---|---|
| *Anthropometric* | | | | |
| Height (cm) | 420,910 | −0.1680 | 0.0612 | 0.006 |
| Weight (kg) | 420,697 | −0.0644 | 0.1361 | 0.636 |
| BMI (kg/m$^2$) | 420,623 | 0.0326 | 0.0459 | 0.477 |
| *Cardiovascular risk factors* | | | | |
| DBP (mmHg) | 421,799 | −0.8240 | 0.0791 | $2.0 \times 10^{-25}$ |
| SBP (mmHg) | 421,797 | 0.0760 | 0.1560 | 0.626 |
| Pulse pressure | 421,797 | 0.9000 | 0.1140 | $3.0 \times 10^{-15}$ |
| Mean arterial pressure | 421,797 | −0.5240 | 0.0969 | $6.4 \times 10^{-8}$ |
| Hypertension | 422,334(33.85%) | 0.925 | 0.888–0.964 | $2.3 \times 10^{-4}$ |
| Coronary artery disease | 422,334(7.48%) | 1.022 | 0.950–1.100 | 0.554 |
| Atrial fibrillation | 422,334(3.71%) | 1.071 | 0.969–1.184 | 0.178 |
| Ventricular arrhythmia | 422,334(0.56%) | 0.868 | 0.674–1.117 | 0.271 |
| Diabetes Mellitus | 422,334(7.04%) | 1.072 | 0.996–1.155 | 0.064 |
| *Other* | | | | |
| Cancer (malignant) | 422,334(15.35%) | 0.983 | 0.932–1.036 | 0.512 |
| Depression | 422,334(14.35%) | 1.041 | 0.986–1.098 | 0.144 |
| Reaction time (ms) | 417,771 | −0.7016 | 1.0544 | 0.506 |
| Fluid intelligence score | 105,348 | −0.0645 | 0.0398 | 0.106 |
| Parental lifespan | 158,649 | 0.0792 | 0.0229 | $5.5 \times 10^{-4}$ |

The effect of the polygenic score of heart-rate (HR) response to exercise on cardiovascular and non-cardiovascular phenotypes in the UK Biobank cohort was performed in participants that were not part of the discovery GWAS. Effect sizes are shown as the incremental change in phenotype for continuous phenotypes or as odds ratio for binary traits, for one unit change in polygenic score. Every unit change in polygenic risk corresponds to one standard deviation change in HR response to exercise. Supplementary Table 12 shows the effect estimates per phenotype of HR response

found that genes involved in the sinoatrial node were enriched[22]. The sinoatrial node genes *GNG11* and *RGS6* that have been both previously associated with HR variability[22] were chiefly associated with HR variability in this study as well. This finding emphasizes that it is important for follow-up studies to focus on extracting more different HR phenotypes before, during, and after exercise. These phenotypes should be jointly analyzed to further increase the resolution of HR-specific SNP associations. Currently the ability to replicate these findings in external cohorts is limited due a lack of available data concerning both HR profile during exercise and genetics. Opportunities for larger studies of HR recovery and HR increase may occur in the future as more and larger biobanks become available.

Observational studies have demonstrated strong associations of HR recovery and HR increase with sudden cardiac death, all-cause death, cardiovascular death[4,5,24], and even cancer[27]. These studies all suggest that autonomic impairment, the imbalance of vagal and adrenergic tone, increases susceptibility to diseases, mortality, and life-threatening arrhythmias. In the current study, we observed that a genetically increased HR recovery and HR increase was significantly associated with higher parental age, but not with ventricular arrhythmia, atrial fibrillation, or other diseases and phenotypes. Since the polygenic risk score was not significantly associated with mother's age of death, we could not reliably establish a true-positive association with parental age. Although, the notion that life-threatening arrhythmia's occur more often in men than in women could explain this discrepancy[43]. Regardless, whether or not the association is a true-positive one, it is possible to conclude from our results that HR response to exercise may not be as important for the human lifespan as other more established risk factors such as blood pressure, lipids, BMI, or educational attainment[44]. The association with parental age should be examined in follow-up studies with independent cohorts, but statistical power may be difficult realize given the exceptionally large sample size of this study. Future Mendelian randomization studies should be conducted in even larger cohorts and with other disease outcomes, such as fatal arrhythmias to provide a better understanding of the clinical consequences.

In conclusion, this is a well-powered genetic study of HR recovery and HR increase; we identified 25 genetic signals in 23 loci to be genome-wide significantly associated. This study adds a new line of evidence to the theory that components of the autonomous nervous system are underlying inter-individual differences in HR recovery.

## Methods

**Measurement of the HR profile and quality control.** The UK Biobank is a cohort of individuals with an age range of 40–69 registered with a general practitioner of the UK National Health Service. In total 503,325 individuals were included and provided informed consent between 2006 and 2010. The UK Biobank cohort study was approved by the North West Multi-centre Research Ethics Committee (reference number 06/MRE08/65). Detailed methods used by UK Biobank have been described elsewhere[45].

In total, 99,539 ECG exercise records were taken for 96,567 participants who underwent a cardio assessment; 79,217 were performed during the baseline visit (2006–2010), and 20,322 were performed at the second follow-up visit (2012–2013). The participants were asked to sit on a stationary bike, start cycling after 15 s of rest, and then perform six minutes of physical activity, after which exercise was terminated and participants sat down for about one minute without cycling. The exercise protocol was adapted according to participants' risk factors; details can be found elsewhere[46]. Participants were only included in the study if they were allowed to cycle at 50% or 30% of their maximum workload (no risk to minimum risk), as described further in the 'Statistical analyses (exclusions)' section. The exercise was ended after participants reached a pre-set maximum HR level of 75% of their age-predicted maximum HR. The cardio assessment involved a 3 lead (lead I, II and III) ECG recording (AM-USB 6.5, Cardiosoft v6.51) at a frequency of 500 Hz. The ECG was recorded using four electrodes placed on the right and left antecubital fossa and wrist and stored in an xml-file of Cardiosoft.

Of all available ECG records, 77,190 contained full disclosure data that could be used to detect R waves; other records contained an error relating to the ECG device used ('*Error reading file C:/DOCUME~1/UKBBUser/LOCALS~1/Temp/ONL2F. tmp*'). R waves were detected with the gqrs algorithm[47] and further processed using Construe[48] (https://github.com/citiususc/construe) to detect individual Q-R-S waves. Following international recommendations to obtain reliable RR intervals[49], abnormal values (0.286–2 s) were removed. Additional outliers were removed using the tsclean function, a part of R-package forecast v7.3 that incorporates the method described by Chen and Liu[50] for automatic detection of outliers in time series. A total of 2,804 ECGs were excluded due to excess noise (identified by determining the standard deviation over a rolling standard deviation with a window length of three beats over RR intervals per ECG per phase and removing the 98th percentile of this distribution). In total we inspected about 10,000 RR interval profiles or ECGs to evaluate the RR-interval detection and ensure quality control. For each

ECG, we estimated the mean resting HR, standard deviation of RR intervals (SDNN, log2 transformed), and root mean square of successive differences between RR intervals (RMSSD, log2 transformed) from the RR intervals before exercise started. HR increase was determined as the difference between peak HR during exercise and resting HR. HR recovery was defined as the difference between maximum HR during exercise and mean HR at $10 \pm 3$, $20 \pm 3$, $30 \pm 3$, $40 \pm$, and $50 \pm 3$ s after exercise cessation (HRR10–HRR50). HR recovery at exactly one minute was not available; only nine participants recovered after a duration $\geq 60$ s. Observations of the second follow-up visits were used when no baseline observation was available. Variables were inspected for normality, and participants with extreme ECG exercise measurements (more than $\pm 5$ standard deviations from mean) were excluded on a per-phenotype basis.

By means of external validation, we estimated that resting HR, SDNN, and RMSSD were highly consistent with previous GWAS estimates[21,22]. To this end, we performed linear regressions between the HR traits and their polygenic score (please also see the 'polygenic score' method section). The beta coefficients ($\beta$) of resting HR ($\beta = 1.085$, se = 0.029, $p = 3 \times 10^{-309}$), SDNN ($\beta = 1.145$, se = 0.051, $p = 1 \times 10^{-108}$), and RMSSD ($\beta = 1.0816$, se = 0.043, $p = 2 \times 10^{-139}$) was close to 1 and highly significant.

For the current analyses, HR phenotypes were rank-based inverse normal transformed to increase the power to detect low-frequency variants and allow for comparisons of beta coefficients between traits. Source code, example data, and further descriptions of the methods are available at https://github.com/niekverw/E-ECG.

Individual data on disease prevalence and incidence were obtained from the Assessment Centre in-patient health episode statistics (HES) and self-reports during any of the visits obtained through questionnaires and nurse-interviews, as described previously[51]. Mothers, fathers, and parental age of death were defined according to Pilling et al.'s[44] study; in short, participants aged between 55–70 years were included, only if fathers died at $\geq 46$ years of age or mothers died at $\geq 55$ years of age. If an age of death was missing, questionnaires of follow-up visits were used where available. The lifespan of mothers and fathers were combined into a single normalized parental lifespan. Parental lifespan, as a proxy for mortality, was defined as the primary outcome variable.

**Genotyping and imputation**. Genotyping, quality control, and imputation to three reference panels (HRC v1.1,1000 genome and UK10K) was performed by The Wellcome Trust Centre for Human Genetics, as described in detail elsewhere[52]. Sample outliers (based on heterozygosity or missingness) were excluded, and 373 participants were excluded on the basis of gender mismatches. The analyses were restricted to SNPs of the HRC v1.1 imputation panel. Post-GWAS analyses were conducted using SNPs with a minor allele frequency greater than 1% and an imputation quality score of more than 0.3. Summary statistics deposited online will include all SNPs.

**Statistical analysis**. Regression analyses of resting HR, SDNN, and RMSSD were adjusted for gender, age, gender-age interaction, body mass index (BMI), BMI*BMI, the first 30 principal components, and genotyping chip (Affymetrix UK Biobank Axiom or Affymetrix UK BiLEVE Axiom array). To fully account for aerobic exercise capacity in HR increase and HR recovery, the model also included exercise duration, exercise program (30% or 50% maximum load), maximum workload achieved, and the interaction between exercise program and maximum workload achieved.

Participants were excluded if they stopped exercising earlier than planned, experienced chest-pain or other discomfort, were at medium-to-high cardiovascular risk[46] at the time of the test, or terminated exercise for unknown reasons. In a post-hoc analysis, the population was stratified by participants that reported taking sotalol medication, beta-blockers, anti-depressants, atropine, glycosides or other anti-cholinergic drugs, or were previously diagnosed with myocardial infarction, supraventricular tachycardia, bundle branch block, heart failure, cardiomyopathy, or previously had a pacemaker or ICD implant. In a post-hoc sensitivity analysis, the differences in beta estimates in participants with and without cardiovascular disease or HR-altering medication were assessed using a Chow test.

In total, 58,818 participants were included in the GWAS. The genome-wide association study and heritability analyses were performed using BOLT-LMM[53] and BOLT-REML[54], respectively. A conjugate gradient-based iterative framework for fast mixed-model computations was employed to accurately account for population structure and relatedness; additive effects were assumed. The BOLT software was used with 509,255 genotyped SNPs that were extracted from the final imputation set (to ensure a 100% call rate per SNP). After pruning ($R^2 > 0.5$, using plink '--indep-pairwise 50 5 0.5), LD scores also used by BOLT, were estimated from 2,000 randomly selected UK Biobank participants (who were selected after sample exclusions based on relatedness, missingness, and heterozygosity). To control for relatedness among participants in linear logistic, or cox regression analyses, we used cluster-robust standard errors with genetic family IDs as clusters. A family ID was given to individuals belonging together based on $3^{rd}$-degree or closer as indicated by the kinship matrix, which was provided by UK Biobank (kinship coefficient > 0.0442). All statistical analyses other than the genome-wide analysis were carried out using R v3.3.2 or STATA/SE release 13.

Since the current study is by far the largest population-based study of electrocardiographic exercise tests, independent cohorts that matched this study in size and availability of variables (specific HR response variables and genetics) were unavailable for replication purposes. Therefore, a conservative genome-wide significant threshold of $p < 8.3 \times 10^{-9}$ was adopted to account for six independent traits, in accordance with similar multi-phenotype studies of this scale[55–59].

Variants were considered to be independent if the pairwise LD ($R^2$) was less than 0.01. A locus was defined as the highest associated independent SNP $+/-$ 1MB. The strongest associated variant within a locus was assigned the sentinel SNP. If there was evidence for multiple independent SNPs in one locus based on LD, it was confirmed by using linear regression and adjusting for the sentinel SNP.

**Pleiotropy analyses**. The GWAS catalog database (https://www.ebi.ac.uk/gwas/) was queried by searching for SNPs in a 1MB distance of the SNPs found in this study. LD was determined by calculating the $R^2$ and $D'$ in the UK Biobank between the GWAS catalog SNPs and the SNPs found in this study. In addition we examined genome-wide summary statistics for 699 traits using PhenoScanner[23] (v1.1, http://www.ner.medschl.cam.ac.uk/phenoscanner). PhenoScanner was used to cross-reference HRR associated SNPs for their association with a broad range of phenotypes regardless of genome-wide significance.

To gain insights into pleiotropy among HR variables, we performed linear regression analyses for all significantly associated SNPs with resting HR, HR variability (SDNN and RMSSD), HR increase, and HR recovery. The Z-scores, which were aligned to the HR recovery increasing allele, were visualized in a heat plot.

**Polygenic score**. Polygenic scores of HR increase and HR recovery were constructed by calculating the sum of the number of alleles that increased HR increase or HR recovery weighted by the corresponding beta coefficients. The primary polygenic score was based on all primary and secondary SNPs that were genome-wide significantly associated. The relationships between the polygenic score and clinical phenotypes were tested in 422,947 individuals who were not part of the discovery GWAS, using linear, logistic, and cox regression analyses. The discovery sample was excluded from this analysis to avoid any potential bias or reverse confounding. The statistical power for a case-control Mendelian randomization in this study ($N = 422,334$) was calculated at $\alpha = 0.05$ using the sample size, proportion of cases, strength of the polygenic risk score, and the expected causal hazards ratio[60].

**Functional variants and candidate causal genes**. To search for evidence of the functional effects of HR profile associated SNPs, we used multiple QTL databases including the following: Stockholm–Tartu Atherosclerosis Reverse Network Engineering Task (STARNET)[61], GTEX, version 6[62], cis-eQTL datasets of Blood[63–65], and cis-meQTLs[66]. Only eQTLS/meQTLs that achieved $p < 1 \times 10^{-6}$ and were in LD ($R^2 > 0.8$) with the queried SNP were considered significant.

Long-range chromatin interactions with the 1MB region at either side of a SNP were examined and visualized using HUGIn[67]. Only genes that achieved a Bonferonni significant association and demonstrated a clear pattern of interaction between the queried SNP and the promoter region were prioritized.

For all primary and secondary SNPs that were genome-wide significantly associated, candidate causal genes were prioritized as follows: a) by proximity, the nearest gene or any gene within 10 kb; b) by protein-coding gene variants in LD ($R^2 > 0.8$); c) by eQTL analysis (described above); and d) by long-range chromatin interaction analysis (described above).

**Data availability**. The data that support the findings of this study are available from the corresponding author upon reasonable request. The de novo GWAS analysis (complete summary statistics of all genetic variants and traits) have been deposited in Mendeley with the identifier 'doi:10.17632/tg5tvgm436.1'.

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

## Acknowledgements

This research has been conducted using the UK Biobank Resource. We would like to thank the Center for Information Technology of the University of Groningen for their support and for providing access to the Peregrine high performance-computing cluster. We also thank Dr. Thomas Teijeiro for his assistance with the Construe algorithm. N. Verweij is supported by NWO VENI (016.186.125), which was awarded to study the mechanisms underlying electrocardiographic changes in response to exercise, and by a Marie Sklodowska-Curie GF (call: H2020-MSCA-IF-2014, Project ID: 661395).

## Author contributions

N.V. designed the study and supervised analyses. N.V., Y.J.v.d.V., and P.v.d.H. played a role in data collection, analyses, and drafting the manuscript.

## Additional information

**Competing interests:** The authors declare no competing interests.

