## [Peer Review File · Nature Communications]

Reviewers' comments:

Reviewer #1 (Remarks to the Author):

The authors have attempted to address a challenging question in this study. After conducting a thorough analysis of the phenotype, the authors conclude that the HR response to exercise is modulated by regions of the genome which have a role in the autonomous nervous system. They refute the idea of causal relationship with mortality or cardiovascular disease but needs further study. I find this re-assuring that autonomous nervous system came up in pathway analysis and a bit disconcerting that the polygenic score did not show strong association with other cardiovascular diseases.

1. I would like to see if the authors used a more "inclusive" list of SNPs using a FDR approach and did their downstream pathway and GRS analysis to get some more power. (Maybe just a primary polygenic risk score against parental lifespan & Hypertension)
2. Also, it will be interesting to see if the cardiovascular risk stays the same or increases if you test your GRS (either using GWAS SNPs or FDR SNPs) in the remaining UKBB samples by splitting them in exercisers/non-exercisers.
3. As GWAS catalog only stores results before a certain p-value threshold and also in studies which are "GWAS", I wondered if you tried Phenoscanner software to look for pleiotropy.
4. The URL on line 272 is a dead link.
5. Line 217: I tend to disagree with the statement: "Multiple eQTLs observed at the same SNP ... true eQTL." A SNP affecting one gene could be equally true. Alternatively, one SNP affecting many genes might also suggest that the SNP/gene association is not disease causing and is probably housekeeping?
6. Line 107: Not sure I understand the error? Is this relevant?

Reviewer #2 (Remarks to the Author):

This study presents the largest genetic study to date on measures of heart-rate profile during exercise. It will substantially add to the literature in this area. I have a few comments below. If addressed then I would highly recommend for publication.

Comments:

1. While overall very well written, there are some awkward phrases and incorrect terms/work usage. I would recommend having the English reviewed in the final version. I have listed some particular points below.
 - a) L79 - consider replacing "determinants" with "variants"
 - b) L178 - 3th should be 3rd
 - c) L205 - "increases" should be "increase" & "weighted for" should be "weighted by"
 - d) L219-222 - awkward phrasing..consider "Long range chromatin interactions with the 1MB region either side of each sentinel SNP were examined and visualized using HUGin. Only genes that achieved a Bonferroni significant association and demonstrated a clear pattern of interaction between the sentinel SNP and the promoter region were prioritized."
 - e)L239 - change to "SDNN and RMSSD were included for comparison."
 - f) L255 - don't need "in history"
 - g) L256 - "we make available" should be "we have made available"
 - h) L262 - consider changing to "prioritized based on proximity to the sentinel SNP"
 - i) L266 - consider changing to "Several genes had multiple lines of evidence (shown in Table 1), ..."
 - j) L271 - "tissues IDENTIFIED after"
 - k) L296 - you mean "disentangle"
 - l)L307 - I think you mean "for HR-increase" not "of HR-increase"
2. Please check that any decimal numbers use dots rather than commas (ie L110)

3. In the Methods section where you describe the UK Biobank cohort (p4), it would be useful to describe the ethnic makeup of the population, particularly those in your subset. There are some non-European individuals in UKBB so want to make sure that they have either been excluded from your sample or correctly accounted for in the genetic analysis.
4. When referring to "rank-based inverse normal" transformations the "rank-based" is redundant so could be removed.
5. L188-189 - it is confusing to refer to your locus as 1MB region surrounding the SNP. Please indicate if this is SNP +/- 500kb or SNP +/- 1MB.
6. L205 - it is redundant to include "multiplied" after weighted since it is understood in the field that this is what you did.
7. In the methods section on "Functional variants and candidate genes" please indicate if only the sentinel SNPs were queried or if you also included the secondary signals (ie all significant independent SNPs).
8. Please include SE for your heritability estimates if they are available from the program (in text and Fig 1)
9. L257-258 - please specify what other "open access repositories" will be used and provide links in the final copy so that readers can easily find. Please ensure that the summary stats are posted to "cardiomics.net" as soon as the paper goes live online.
10. For the pathway analysis in DEPICT, I am assuming the enrichment analysis is done within in the program. How does it select the control set of genes? Probably a short description of the enrichment procedure should be added to the methods section.
11. L309 - were the polygenic scores created using the published GWAS data for disease outcomes or for your HR traits? If for the HR traits, did you include only your sentinel SNPs or all independent SNPs?
12. L318-321 - this sentence is confusing and not sure what you are trying to say. Your study used UKBB data so if you just mean in your study then please remove the reference to the study. If you meant something else then please rephrase to make clearer.
13. L344 - please replace "candidate genes" with "associated genes" since candidate genes sounds like you have pulled the genes from the literature rather than doing a GWAS study.
14. L415-417 - There are more and more large biobanks coming online, so while I agree that it is not possible for you to replicate your findings at this time, there may be an opportunity for larger studies in the near future (if they measure this phenotype). Another opportunity may be to look at more specific phenotype subtypes to give more insight into the biology, which could be done in smaller populations. Also, looking at these phenotype in other ethnic groups could leverage different LD structures to pinpoint causal genes. These are all suggestions for wording and future studies, not expectations for inclusion in this paper.
15. The standard in the field is "genome-wide association study" rather than "genome wide association study" so please correct throughout.

Tables and Figures

Table 1

- i) Please include chr:position for these SNPs rather than "region" as it makes it a lot easier for readers to quickly align your findings with their own, especially since a lot of the imputation panels are using these as identifiers.
- ii) The column header and footnote do not match for the 3rd column. You could also consider removing the NEA from that column if you are worried about space.
- iii) Please split the beta(se) column or adding more spacing. You could also consider using less significant digits for the SE.

Figure 3 - are these units correct that they are Z-scores? I thought the results of inverse-normal transformations weren't z-scores since they were rank-based so not directly SE? Please double-check and if you have just used the beta's from your analysis then say this.

Supplementary Figures 2 & 3 - Make much bigger since even when you zoom in on the PDFs you can't read the gene names or region info so are not useful as is.

Reviewed by - Dr Jennifer E Huffman

Point-by-point response to reviewers

Reviewer #1

The authors have attempted to address a challenging question in this study. After conducting a thorough analysis of the phenotype, the authors conclude that the HR response to exercise is modulated by regions of the genome which have a role in the autonomous nervous system. They refute the idea of causal relationship with mortality or cardiovascular disease but needs further study. I find this re-assuring that autonomous nervous system came up in pathway analysis and a bit disconcerting that the polygenic score did not show strong association with other cardio-vascular diseases.

We greatly appreciate the Reviewer's overall positive assessment of our work.

1. I would like to see if the authors used a more "inclusive" list of SNPs using a FDR approach and did their downstream pathway and GRS analysis to get some more power. (Maybe just a primary polygenic risk score against parental lifespan & Hypertension)

This is a good suggestion. We used SNPs below the genome-wide significance for the downstream pathway analysis using DEPICT, as suggested by the creators (we used the default settings which uses all SNPs $P < 1e-5$). We modified the sentence to reflect this more accurately:

“Pathway analyses were attempted with “DEPICT” using default settings (which uses all SNPs $p < 1 \times 10^{-5}$)” and included the reference Pers et al⁴⁴.

Based on the reviewer suggestion, we also tested a GRS based using sub-threshold SNPs and an FDR threshold (corresponding to the $P < 1 \times 10^{-6}$ threshold) to test whether power was improved. However, this approach lead to less significance of our findings compared to the primary GRS used in the manuscript; the t-statistic dropped from 43.17 to 39.14 for resting HR, similarly for blood pressure as well. In literature we found that others also observed that relaxing the genome-wide significant threshold for a GRS does not necessarily lead to an increase in power¹. Since any interpretation of the results generated based on the GRS will not be straightforward, we decided not to include these subsidiary analyses it in this manuscript.

2. Also, it will be interesting to see if the cardiovascular risk stays the same or increases if you test your GRS (either using GWAS SNPs or FDR SNPs) in the remaining UKBB samples by splitting them in exercisers/non-exercisers.

We split the GRS according to the median as suggested by the reviewer and did not observe a difference in in cardiovascular risk (any cardiovascular disease $P=0.420$; coronary artery disease $P = 0.402$; Atrial fibrillation $P= 0.406$). As splitting by the median might be considered arbitrary and is

less powerful than the continuous analyses we did not include this subsidiary analysis in the manuscript.

3. As GWAS catalog only stores results before a certain p-value threshold and also in studies, which are "GWAS", I wondered if you tried Phenoscanner software to look for pleiotropy.

We now included results from the Phenoscanner software in the method section:

"In addition we examined genome-wide summary statistics for 699 traits using PhenoScanner³⁵ (v1.1, <http://www.ner.medschl.cam.ac.uk/phenoscanner>). PhenoScanner was used to cross-reference HRR associated SNPs for their association with a broad range of phenotypes regardless of genome-wide significance. "

In the results:

"PhenoScanner³⁵ also indicated that HR profile SNPs had pleiotropic effects with resting heart rate, atrial fibrillation, and other electrocardiographic traits (Supplementary Table 11)."

4. The URL on line 272 is a dead link.

Thanks very much. We notified Prof dr. L. Franke, the developer and hoster, who is working on getting it online again. This is an issue that also affects many other publications (e.g. ²⁻⁶) and websites to the same ip (<http://www.genenetwork.nl/genenetwork/> and www.prioritizer.nl), we expect the host to be up again soon. We have added the literature reference of GeneNetwork⁴⁵ that describes the method so that it is not dependent on the URL.

5. Line 217: I tend to disagree with the statement: "Multiple eQTLs observed at the same SNP ... true eQTL." A SNP affecting one gene could be equally true. Alternatively, one SNP affecting many genes might also suggest that the SNP/gene association is not disease causing and is probably housekeeping?

We agree, we removed this sentence.

6. Line 107: Not sure I understand the error? Is this relevant?

Yes we believe that this is very relevant. Because the UK Biobank contains around 99 thousand recorded ECGs (this summary information is available online to anyone), a reader may wonder why

only a proportion, about 2/3th, of the ECGs were analyzed in this study. Since we are the first to have analyzed it, it may be useful, and a reference to anyone interested in exercise ECGs.

Reviewer #2

This study presents the largest genetic study to date on measures of heart-rate profile during exercise. It will substantially add to the literature in this area. I have a few comments below. If addressed then I would highly recommend for publication.

Comments:

1. While overall very well written, there are some awkward phrases and incorrect terms/work usage. I would recommend having the English reviewed in the final version. I have listed some particular points below.

We sincerely thank dr. Huffman for carefully reviewing our paper, the encouraging comments and kind words. The manuscript has now been reviewed and edited by a native English speaker.

a) L79 - consider replacing "determinants" with "variants"

Indeed, corrected.

b) L178 - 3th should be 3rd

corrected.

c) L205 - "increases" should be "increase" & "weighted for" should be "weighted by"

corrected.

d) L219-222 - awkward phrasing..consider "Long range chromatin interactions with the 1MB

region either side of each sentinel SNP were examined and visualized using HUGin. Only genes that achieved a Bonferroni significant association and demonstrated a clear pattern of interaction between the sentinel SNP and the promoter region were prioritized."

We agree and have changed the sentence accordingly.

e)L239 - change to "SDNN and RMSSD were included for comparison."

Corrected.

f) L255 - don't need "in history"

Corrected.

g) L256 - "we make available" should be "we have made available"

Corrected.

h) L262 - consider changing to "prioritized based on proximity to the sentinel SNP"

Corrected.

i) L266 - consider changing to "Several genes had multiple lines of evidence (shown in Table 1), ..."

Corrected.

j) L271 - "tissues IDENTIFIED after"

Corrected.

k) L296 - you mean "disentangle"

Yes, typo, Corrected.

l) L307 - I think you mean "for HR-increase" not "of HR-increase"

Yes, Corrected.

2. Please check that any decimal numbers use dots rather than commas (ie L110)

Corrected.

3. In the Methods section where you describe the UK Biobank cohort (p4), it would be useful to describe the ethnic makeup of the population, particularly those in your subset. There are some non-European individuals in UKBB so want to make sure that they have either been excluded from your sample or correctly accounted for in the genetic analysis.

We agree, we have updated Supplementary Table 1 with the following information:

Ethnicity	N (%)
White	54137(92.04)
Other/Unknown	1024(1.74)
Asian	1744(2.97)
Black	1445(2.46)
Mixed	468(0.8)

Population stratification was accounted for (1.) by including the first 30 principal components, and (2.) by employing BOLT-LMM^{7,8} for the GWAS analysis, which further efficiently controls for population structure; as also can be concluded from the QQ-plots and LD-score estimates (**Supplementary Figure 1**).

4. When referring to "rank-based inverse normal" transformations the "rank-based" is redundant so could be removed.

We agree that these terms have been used inter-changeably in literature. We believe "rank-based inverse normal transformation" of phenotypes is more accurate since "inverse normal transformation" does not mention whether it is *rank* or *non-rank* based and is widely used in literature⁹.

5. L188-189 - it is confusing to refer to your locus as 1MB region surrounding the SNP. Please indicate if this is SNP +/- 500kb or SNP +/- 1MB.

Corrected.

6. L205 - it is redundant to include "multiplied" after weighted since it is understood in the field that this is what you did.

Ok, corrected.

7. In the methods section on "Functional variants and candidate genes" please indicate if only the sentinel SNPs were queried or if you also included the secondary signals (ie all significant independent SNPs).

We prioritized genes for all primary and secondary signals that were genome wide significant. The following sentence in the methods-section was modified: "*For all primary and secondary SNPs that were genome-wide significantly associated, candidate causal genes were prioritized as follows: (...)*"

8. Please include SE for your heritability estimates if they are available from the program (in text and Fig 1)

Corrected.

9. L257-258 - please specify what other "open access repositories" will be used and provide links in the final copy so that readers can easily find. Please ensure that the summary stats are posted to "cardiomics.net" as soon as the paper goes live online.

We modified the sentence to: *"To facilitate future studies, complete summary statistics of all genetic variants and traits can be downloaded from <http://dx.doi.org/10.17632/tg5tvgm436.1>"*

Data is already deposited at Mendeley under <http://dx.doi.org/10.17632/tg5tvgm436.1>, which will be live soon after we received a DOI for this article so that we can reference it properly.

10. For the pathway analysis in DEPICT, I am assuming the enrichment analysis is done within in the program. How does it select the control set of genes? Probably a short description of the enrichment procedure should be added to the methods section.

The DEPICT software projects a membership score to all genes belonging to a certain biological pathway (reconstituted gene-sets) – all genes will have a score for all pathways – this is used to calculate the enrichment in a series of steps. We believe the algorithm is best described in the original paper of *Pers et al.*¹⁰, which we reference.

We modified the following sentence in the results to:

*"Pathway analyses were attempted with "DEPICT" using default settings (which uses all SNPs $p < 1 \times 10^{-5}$), a tool that can prioritize genes, pathways, and tissues by using the genomic region surrounding SNPs as input (please see *Pers et al.*⁴⁴ for a detailed description of the methods)."*

11. L309 - were the polygenic scores created using the published GWAS data for disease outcomes or for your HR traits? If for the HR traits, did you include only your sentinel SNPs or all independent SNPs?

Both primary and secondary SNPs were used. We included the following sentence *"The primary polygenic score was based on all primary and secondary SNPs that were genome-wide associated."*

12. L318-321 - this sentence is confusing and not sure what you are trying to say. Your study used UKBB data so if you just mean in your study then please remove the reference to the study. If you meant something else then please rephrase to make clearer.

We agree, we removed the reference and modified the sentence to “*The statistical power for a case/control Mendelian randomization in this study (N=422,334) was calculated at alpha=0.05, as described previously*³⁵”

13. L344 - please replace "candidate genes" with "associated genes" since candidate genes sounds like you have pulled the genes from the literature rather than doing a GWAS study.

Indeed we do not refer to genes identified in ‘candidate gene studies’ or other literature. The term “candidate genes” and similar terms like “candidate target genes” and “candidate causal genes” in similar context have been used previously in many papers (e.g. ¹¹⁻¹⁵, ¹⁶ and ¹⁷) to refer to the most probable causal genes at GWAS-loci. We have replaced “candidate genes” with “candidate causal genes” to be more specific.

14. L415-417 - There are more and more large biobanks coming online, so while I agree that it is not possible for you to replicate your findings at this time, there may be an opportunity for larger studies in the near future (if they measure this phenotype). Another opportunity may be to look at more specific phenotype subtypes to give more insight into the biology, which could be done in smaller populations. Also, looking at these phenotype in other ethnic groups could leverage different LD structures to pinpoint causal genes. These are all suggestions for wording and future studies, not expectations for inclusion in this paper.

We thank the reviewer for realizing this, we concur. However, leveraging different LD structures might bring us closer to the causal variant underlying each signal, but not the candidate causal genes per se. We believe that we have discussed the opportunity to include more phenotypes in paragraph 4 of the discussion and extended it with the following:

“Currently the ability to replicate these findings in external cohorts is limited due a lack of available data concerning both HR profile during exercise and genetics. Opportunities for larger studies of HR-recovery and HR-increase may occur in the future as more and larger biobanks become available.”

Also added:

“Data of tissues that are relevant for the (para-)sympathetic nervous system of the heart is limited, which makes it difficult to dissect molecular mechanisms. Future research should pursue functional follow-up studies of the genetic loci presented here, to pinpoint causal variants, genes and biological mechanisms underlying HR-profile during exercise.”

15. The standard in the field is "genome-wide association study" rather than "genome wide association study" so please correct throughout.

This is now corrected throughout the manuscript.

Tables and Figures

Table 1

i) Please include chr:position for these SNPs rather than "region" as it makes it a lot easier for readers to quickly align your findings with their own, especially since a lot of the imputation panels are using these as identifiers.

We have now included the chromosome and position and removed the 'region'-column (which is still in Supplementary Table 2).

ii) The column header and footnote do not match for the 3rd column. You could also consider removing the NEA from that column if you are worried about space.

Thanks for spotting this, corrected.

iii) Please split the beta(se) column or adding more spacing. You could also consider using less significant digits for the SE.

Corrected, the standard error is now in a separate column.

Figure 3 - are these units correct that they are Z-scores? I thought the results of inverse-normal transformations weren't z-scores since they were rank-based so not directly SE? Please double-check and if you have just used the beta's from your analysis then say this.

Yes these are the Z-scores (= beta divided by SE) for each SNP-phenotype association, which is a measure of significance and direction of effect (similar to the signed-P value of each association – which would give the same information).

Supplementary Figures 2 & 3 - Make much bigger since even when you zoom in on the PDFs you can't read the gene names or region info so are not useful as is.

Yes, we noticed this too. This is because Nat Commun requires word files as supplementary files. We have now added these as high-quality datasets and will provide them at <http://dx.doi.org/10.17632/tg5tvgm436.1> when necessary.

Reviewed by - Dr Jennifer E Huffman

Literature

1. Rosenberg, M. A. *et al.* Validation of Polygenic Scores for QT Interval in Clinical Populations. *CLINICAL PERSPECTIVE. Circ. Cardiovasc. Genet.* **10**, e001724 (2017).
2. Jankipersadsing, S. A. *et al.* A GWAS meta-analysis suggests roles for xenobiotic metabolism and ion channel activity in the biology of stool frequency. *Gut* **66**, 756–758 (2017).
3. Xu, C.-J. *et al.* The emerging landscape of dynamic DNA methylation in early childhood. *BMC Genomics* **18**, (2017).
4. Ferreira, M. A. *et al.* Shared genetic origin of asthma, hay fever and eczema elucidates allergic disease biology. *Nat. Genet.* **49**, 1752 (2017).
5. Vojinovic, D. *et al.* Variants in *TTC25* affect autistic trait in patients with autism spectrum disorder and general population. *Eur. J. Hum. Genet.* **25**, 982 (2017).
6. Nibbeling, E. A. R. *et al.* Using the shared genetics of dystonia and ataxia to unravel their pathogenesis. *Neurosci. Biobehav. Rev.* **75**, 22–39 (2017).
7. Loh, P.-R. *et al.* Efficient Bayesian mixed-model analysis increases association power in large cohorts. *Nat. Genet.* **47**, 284–290 (2015).
8. Loh, P.-R., Kichaev, G., Gazal, S., Schoech, A. P. & Price, A. L. Mixed model association for biobank-scale data sets. *bioRxiv* 194944 (2017). doi:10.1101/194944
9. Beasley, T. M., Erickson, S. & Allison, D. B. Rank-Based Inverse Normal Transformations are Increasingly Used, But are They Merited? *Behav. Genet.* **39**, 580–595 (2009).
10. Pers, T. H. *et al.* Biological interpretation of genome-wide association studies using predicted gene functions. *Nat. Commun.* **6**, 5890 (2015).
11. Hart, A. *et al.* Association analyses identify 38 susceptibility loci for inflammatory bowel disease and highlight shared genetic risk across populations. *Nat. Genet.* **47**, 979 (2015).
12. van der Harst, P. *et al.* Seventy-five genetic loci influencing the human red blood cell. *Nature* **492**, 369–375 (2012).
13. Global Lipids Genetics Consortium. Discovery and refinement of loci associated with lipid levels. *Nat. Genet.* **45**, 1274–1283 (2013).
14. Chambers, J. C. *et al.* Genome-wide association study identifies loci influencing concentrations of liver enzymes in plasma. *Nat. Genet.* **43**, 1131–1138 (2011).
15. Teslovich, T. M. *et al.* Biological, clinical and population relevance of 95 loci for blood lipids. *Nature* **466**, 707–713 (2010).
16. Eliassen, A. H. *et al.* Association analysis identifies 65 new breast cancer risk loci. *Nature* **551**, 92 (2017).
17. Dunning, A. M. *et al.* Identification of 19 new risk loci and potential regulatory mechanisms influencing susceptibility to testicular germ cell tumor. *Nat. Genet.* **49**, 1133 (2017).

REVIEWERS' COMMENTS:

Reviewer #1 had no further comments to the authors

Reviewer #2 (Remarks to the Author):

The authors have addresses all my comments and suggestions to my satisfaction and would recommend this manuscript be accepted for publication.

To address the editor's concern, I agree with the authors about the possibility of replication at this time. Due to the very large discovery sample size, the lack of an appreciable sized sample available for replication in existence at this time, and the author's conservative p-value threshold for defining a locus as genome-wide significant, I am fine with accepting this manuscript for publication without providing replication.

I noticed 2 very minor things in the Supplementary Tables. In ST9 & 13, there didn't appear to be a description or legend of what the colors mean. Also, it looks like the authors did not updated the numbering or the index after adding in ST11.

Reviewed by Dr Jennifer E Huffman

Point-by-point response to reviewers

Reviewer #1 had no further comments to the authors

Reviewer #2 (Remarks to the Author):

The authors have addresses all my comments and suggestions to my satisfaction and would recommend this manuscript be accepted for publication.

To address the editor's concern, I agree with the authors about the possibility of replication at this time. Due to the very large discovery sample size, the lack of an appreciable sized sample available for replication in existence at this time, and the author's conservative p-value threshold for defining a locus as genome-wide significant, I am fine with accepting this manuscript for publication without providing replication.

I noticed 2 very minor things in the Supplementary Tables. In ST9 & 13, there didn't appear to be a description or legend of what the colors mean. Also, it looks like the authors did not updated the numbering or the index after adding in ST11.

The supplementary tables have been updated according to the final remarks of reviewer 2. We'd like to thank both reviewers for their review work and comments.